# Effects of Perineal Warm Compresses during the Second Stage of Labor on Reducing Perineal Trauma and Relieving Postpartum Perineal Pain in Primiparous Women: A Systematic Review and Meta-Analyses

**DOI:** 10.3390/healthcare12070702

**Published:** 2024-03-22

**Authors:** Ruiyang Sun, Jing Huang, Xiu Zhu, Rui Hou, Yu Zang, Yuxuan Li, Jingyu Pan, Hong Lu

**Affiliations:** 1School of Nursing, Beijing University of Chinese Medicine, Beijing 102488, China; sunruiyang@bucm.edu.cn; 2Division of Care for Long Term Conditions, Florence Nightingale Faculty of Nursing, Midwifery and Palliative Care, King’s College London, London SE1 8WA, UK; jing.huang@kcl.ac.uk; 3School of Nursing, Peking University, Beijing 100191, China; zhuxiu@bjmu.edu.cn (X.Z.); hourui@bjmu.edu.cn (R.H.); pjy8116@bjmu.edu.cn (J.P.); 4School of Nursing, Hebei Medical University, Shijiazhuang 050017, China; 18701186@hebmu.edu.cn; 5Vanke School of Public Health, Tsinghua University, Beijing 100084, China; li-yx23@mails.tsinghua.edu.cn

**Keywords:** warm compresses, primiparity, perineal trauma, delivery

## Abstract

Non-pharmaceutical midwifery techniques, including perineal warm compresses, to improve maternal outcomes remain controversial. The aims of this study are to assess the effects of perineal warm compresses on reducing perineal trauma and postpartum perineal pain relief. This systematic review included randomized controlled trials (RCTs). We searched seven bibliographic databases, three RCT register websites, and two dissertation databases for publications from inception to 15 March 2023. Chinese and English publications were included. Two independent reviewers conducted the risk of bias assessment, data extraction, and the evaluation of the certainty of the evidence utilizing the Cochrane risk of bias 2.0 assessment criteria, the Review Manager 5.4, and the online GRADEpro tool, respectively. Seven RCTs involving 1362 primiparous women were included. The combined results demonstrated a statistically significant reduction in the second-, third- and/or fourth- degree perineal lacerations, the incidence of episiotomy, and the relief of the short-term perineal pain postpartum (within two days). There was a potential favorable effect on improving the integrity of the perineum. However, the results did not show a statistically significant supportive effect on reducing first-degree perineal lacerations and the rate of perineal lacerations requiring sutures. In summary, perineal warm compresses effectively reduced the second-, third-/or fourth-degree perineal trauma and decreased the short-term perineal pain after birth.

## 1. Introduction

Perineal laceration, which significantly affected the new mother’s physical, psychological, and social experiences toward natural delivery [1], can occur naturally during spontaneous delivery or through episiotomy. In a normal delivery, the incidence of natural perineal laceration and the rate of episiotomy among nulliparous women was 3 times [2] and 2.5 times [3] higher than among parous women, respectively. Specifically, perineal trauma following birth can cause pain, which is usually overlooked by women and their caregivers who may not recognize it as a health problem [4]. Incontinence of urine and feces [5] and sexual dysfunction [6], which are also neglected and not of concern during perinatal or postnatal period by midwives or women themselves [7,8].These issues can lead to reduced engagement in baby care activities and other daily activities of women [5]. Additionally, research findings have shown that the physical symptoms caused by perineal trauma could significantly influence the women’s postnatal psychological symptoms [9,10]. For instance, it may result in postpartum anxiety, post-traumatic stress or depression [11]. Socially, these issues then impact the mother–baby attachments [10], marital relationships [7] and the overall birth experience. This resulting dissatisfaction can reduce the mother’s confidence in giving birth again [12]. In summary, perineal trauma can lead to a relatively negative birth experience among women, particularly for first-time mothers. Therefore, adequate attention should be given, and clinicians and midwives should provide evidence-based midwifery care to reduce women’s physical and psychological symptoms.

Perineal trauma can be classified into four degrees according the Royal College of Obstetricians & Gynaecologists (RCOG) [13]: the first degree (perineal mucosal injury); the second degree (perineal injury involving perineal fascia and muscles); the third degree (involving the anal sphincter complex); and the fourth degree (involving the internal, external of the anal sphincter and sphincter epithelium). Perineal trauma can be caused by episiotomy or occur spontaneously. The incidence of the overall perineal trauma and the third- and/or fourth-degree injuries were reported to be 68.8% [14] or ranging from 0.68% to 11%, respectively [15,16,17,18]. Women reported a higher level of pain complaints with severe (third- and/or fourth-degree) perineal tears [19]. Severe perineal tears lead to more complications, especially in the postpartum period. These complications include urinary and fecal incontinence, sexual dysfunctions, and sex-related health problems [20]. Therefore, minimizing the incidence of severe perineal tears is crucial to improving the positive birth experience for women. Nulliparous women are a high-risk group for severe perineal tears [14], with the risk in nulliparous women being 5.32 times higher compared to multiparous women [21]. Consequently, nulliparity is widely recognized as a remarkable risk factor for perineal trauma.

Numerous midwifery techniques [22] have been suggested as effective ways to enhance the positive experience of normal birth such as hands-on/off [23], perineal massage [24], and heat therapy, including warm perineal compresses. Perineal warm compresses refer to placing warm packs in the perineum during the second stage of labor [25] to alleviate the local muscle tension, increase the blood circulation, and thereby improve maternal comfort. Being non-invasive and safe to conduct, they can help improve the integrity of the perineum, reducing the occurrence of severe perineal trauma. Many midwives and women favor these techniques for relieving perineal pain and enhancing women’s comfort [26]. However, the effects of perineal warm compresses remain inconclusive. Some researchers supported it to reduce perineal pain [27], while others doubted its effectiveness in maintaining an intact perineum [28]. Several relevant systematic reviews explored the effects of perineal warm compresses from different perspectives, such as comparing the effects of combined perineal massage and warm compresses [29], investigating different heat therapy effects on the perineum [30] or utilizing outdated data [31]. Thus, this study aims to investigate the effects of perineal warm compresses during the second stage of labor on maternal outcomes in primiparous women, aiming to provide evidence-based support for both the objectively physical effectiveness and women’s experiences of perineal warm compresses. 

## 2. Materials and Methods

This systematic review was conducted in accordance with the Cochrane Handbook for Systematic Reviews of Interventions (Version 6.3, 2022) and reported following the Preferred Reporting Items for Systematic Reviews and Meta-Analyses Statement (Appendix A). The protocol was registered in advance at PROSPERO (CRD42022299398). 

### 2.1. Study Search Process

The search strategy was developed by the research group and the pre-search was performed in PubMed (for English publications) and CNKI (for Chinese publications) before the formal search. Following the Cochrane Handbook guidelines, the study search was conducted in three steps: firstly, seven mainstream academic databases, including PubMed, Embase, CINAHL Plus, Web of Science, China National Knowledge Infrastructure (CNKI), WanFang Data Knowledge Service Platform (WanFang Data), and Chinese Biomedical Literature Service System (SinoMed), were searched. Secondly, we searched three register websites of randomized trials including the World Health Organization International Clinical Trials Registry Platform (WHO ICTRP), Cochrane Central Register of Controlled Trials (CENTRAL), and United States National Library of Medicine ClinicalTrials.gov (ClinicalTrials.gov). Thirdly, two dissertation databases, including ProQuest Dissertations & Theses Database (PQDT) and China Dissertations Database (CDDB), were also searched. Additionally, we manually searched the references of the studies deemed for full-text review. There were no language or study design restrictions, and the search period extended from inception to 15 March 2023. (Appendix A).

### 2.2. Inclusion and Exclusion Criteria 

Participants: Primiparous women older than eighteen years old who had a singleton fetus with the cephalic presentation and anticipated a normal vaginal birth.

Intervention: Warm perineal packs.

Comparison: Midwifery standard care.

Outcomes: Intact perineum, perineal lacerations including first-, second-, third-, and/or fourth-degree lacerations, perineal lacerations requiring sutures, the incidence of episiotomy, pain relief after delivery, and women’s experiences with warm compresses.

Study design: RCTs published in Chinese and English were included. 

### 2.3. Study Selection Process

The preliminary search results were imported into Endnote 20 by one reviewer to remove the duplicates. Then we adopted a three-step study selection process. In the first step, the titles and abstracts were screened to identify relevant publications. In the second step, the full text was examined to include the targeted publications, and the reasons for excluding each study were documented. In the third step, a snowball method was employed to search the listed references of the included study. All selection processes were carried out by two independent reviewers, with any disagreements resolved by the review group.

### 2.4. Quality Assessment and Data Extraction

The quality of the studies was appraised by two independent reviewers using the Cochrane risk of bias (RoB) 2.0 assessment criteria with an Excel tool (https://www.riskofbias.info/ (accessed on 30 May 2023)). The tool comprises five domains: “bias arising from the randomization process”, “bias due to deviations from intended interventions”, “bias due to missing outcome data”, “bias in measurement of the outcome”, and “bias in the selection of the reported result”. The risk of bias in each domain was rated as “low risk”, “some concerns”, and “high risk” according to the built-in bias assessment pathway map. Subsequently, the final overall risk of bias of each included study was assigned to one of these three judgment levels. 

To ensure consistency, a preliminary data extraction exercise was performed among three studies to reach an agreement. Subsequently, two reviewers independently extracted information, including study location, participant details, sample size, detailed intervention (time to start, duration, and time to stop), comparison, and the detailed outcome (measurement, time to collect, and appraiser). Authors were contacted via email for data verification and conversion if necessary.

### 2.5. Data Analyses and the Certainty Appraisal of the Body of Evidence

Meta-analyses were conducted using the Review Manager 5.4. Effect sizes of the dichotomous or continuous data were calculated as the risk ratios (RRs) or the mean differences (MD), respectively. Point estimates and the 95% confidence intervals (CIs) were calculated for each effect size, and a two-tailed *p*-value less than 0.05 was considered statistically significant. The heterogeneity test determined whether to use the fixed-effect model (*p* > 0.10, I^2^ ≤ 50%) or the random-effect model (*p* ≤ 0.10, I^2^ > 50%). Sensitivity analyses were performed utilizing a one-study-out method to assess results reliability. Subgroup analyses categorized perineal lacerations and the pain relief level during different postpartum periods. To evaluate the certainty of the evidence, we used the online GRADEpro Guideline Development Tool (https://gdt.gradepro.org (accessed on 28 June 2023)). This tool provided relatively objective guidance for clinical practice by considering five potential downgrade factors for the highest level of certainty studies, which were all randomized controlled trials. These factors included high risk of bias, inconsistency, indirectness, imprecision, and other considerations such as the high publication bias. Finally, the certainty of the evidence for each outcome was categorized as “very low”, “low”, “moderate”, or “high”. 

## 3. Results

### 3.1. Study Search, Selection Results, and Study Characteristics

A total of 2902 studies were retrieved from databases (n = 2565) and register websites (n = 328), with nine studies identified from citations. After removing duplicates, 1905 records underwent title and abstract screening. Subsequently, 72 records were reviewed in full text and 65 studies were excluded for various reasons (Appendix A). Ultimately, seven publications (Appendix A) were included in this review [25,32,33,34,35,36,37]. Figure 1 presents the study selection process. 

All seven included studies were RCTs, published between 2007 and 2021 in six countries (Australia, Arabia, Egypt, Iran, Turkey, and China). A total of 1362 primiparous women were enrolled in this review, with 683 women receiving perineal warm compresses and the remaining 679 women receiving the midwifery standard care. The performers were researchers (n = 5) or midwives (n = 2). Six studies specified that the warm compress started during the beginning of the second stage of labor. Two further reported that the warm compresses were applied when the baby’s head began to distend the perineum and the woman felt perineal extension. Regarding the duration, two studies recommended continuous application during the second stage of labor. Two suggested application during each uterine contraction, and the remaining three studies reported specific durations ranging from 5 to 15 min. Only three studies mentioned the temperature of the warm packs (ranging from 38 to 44 °C or 50 °C), while six studies indicated the temperature of the container in which packs were placed (45–50 °C). Stopping times were mentioned in all studies: until delivery (n = 4), until the fetal head is crowned (n = 2), or after the 5–15 min intervention. The main maternal outcomes were perineal pain (n = 6), and perineal outcomes such as an intact perineum, perineal trauma, episiotomy, or the need for perineal sutures (n = 6). Additionally, urinary incontinence, pain relief needs, women’s comfort, and perineal swelling were reported by four studies. Table 1 and Appendix A list the characteristics and the extracted data of the included studies.

### 3.2. Risk of Bias Assessment

Three studies were rated as “low risk” and the remaining four were categorized as “some concerns” (Figure 2). Specifically, three studies were assessed as “low risk” in the “randomization process” domain due to the clear descriptions of random sequence generation and allocation concealment. The detailed assessment of the remaining domains is listed in Appendix A.

### 3.3. Effects of Perineal Warm Compresses during Childbirth on Maternal Outcomes

The effects of perineal warm compresses during the second stage of labor on maternal outcomes, including the rate of the intact perineum, the incidence of perineal lacerations, perineal lacerations requiring suture, the incidence of episiotomy, and the level of postpartum perineal pain, are displayed as follows. No included studies investigated the women’s experiences of receiving perineal warm compresses.

#### 3.3.1. Intact Perineum

We employed RRs and the random effect model to assess the effects of perineal warm compresses on improving perineum integrity. The pooled results of six studies, involving 1262 women, indicated a statistically significant effect of warm compresses in maintaining the intact perineum (RR = 3.36, 95%CI: [1.22, 9.27], *p* = 0.02, Appendix A).

#### 3.3.2. Perineal Lacerations

The pooled results of six studies showed a statistically significant effect of warm compresses used during the second stage of labor in reducing the incidence of perineal lacerations (RR = 0.66, 95% CI: [0.54, 0.82], *p* = 0.0001, Figure 3). Interestingly, subgroup analyses demonstrated that perineal warm compresses had no significant influence on reducing the first-degree perineal lacerations (*p* > 0.05, Figure 3). However, they had significant effects on reducing the second-degree (RR = 0.40, 95% CI: [0.27, 0.59], *p* < 0.0001, Figure 3) and the third- and/or fourth-degree perineal lacerations (RR = 0.34, 95% CI: [0.20, 0.57], *p* < 0.0001, Figure 3).

#### 3.3.3. Perineal Lacerations Requiring Suture

Due to heterogeneity, the random effect model was employed to assess the effects of perineal warm compresses on the perineal lacerations requiring sutures. The combined results of three studies involving 977 women demonstrated no significant effect (*p* > 0.05, Appendix A).

#### 3.3.4. Episiotomy

For the evaluation of the effects of perineal warm compresses on reducing the incidence of episiotomy, five studies involving 1184 primiparous women were considered. The fixed-effect model was employed, and the combined results displayed a statistically significant effect (RR = 0.69, 95%CI: [0.58,0.83], *p* < 0.0001, Figure 4).

#### 3.3.5. Postpartum Perineal Pain

The combined results of four studies, involving 1890 primiparous women and assessing the effectiveness of perineal warm compresses on relieving perineal pain, displayed a statistically significant effect (MD = −0.94, 95%CI: [−1.10, −0.77], *p* < 0.00001, Figure 5). Additionally, the subgroup analyses showed its effectiveness in relieving the perineal pain immediately after delivery (MD = −1.71, 95%CI: [−2.20, −1.21], *p* < 0.00001, Figure 5), on the first day after delivery (MD = −1.04, 95%CI: [−1.29, −0.79], *p* <0.00001, Figure 5), and on the second day postpartum (MD = −0.64, 95%CI: [−0.89, −0.39], *p* <0.00001, Figure 5).

### 3.4. Sensitivity Analyses and Certainty of the Body of Evidence

The one-study-out sensitivity analyses indicated that most pooled results did not change significantly, suggesting the reliability of the review’s findings (Appendix A). We manually tracked the funding sources and found that only one study was funded by the medical university [35], two claimed no funding, and the remaining three did not report any funding information. Regarding the certainty of the body of evidence, three outcomes (third- and/or fourth-degree perineal lacerations, the incidence of episiotomy, and perineal pain after two days postpartum) were rated as “moderate” levels. The certainty of six outcomes, including intact perineum, overall perineal lacerations, second-degree lacerations, perineal lacerations requiring suture, the overall postpartum pain, and perineal pain after one day postpartum, were evaluated as “low”. First-degree perineal lacerations and the perineal pain immediately after delivery were graded as “very low”. Furthermore, the reasons for downgrading were suspected publication bias (n = 11), inconsistency (n = 5), risk of bias (n = 3), and imprecision (n = 2). Details of the entire evaluation process for the certainty of the body of evidence are presented in Appendix A. 

## 4. Discussion

### 4.1. Main Findings

This review demonstrated the beneficial aspects of perineal warm compresses during the second stage of labor on perineal-related outcomes among primiparous women. Firstly, perineal warm compresses effectively decreased the rate of second-, third- and/or fourth-degree perineal lacerations and reduced the incidence of episiotomy. Secondly, it significantly alleviated the perineal pain postpartum, including immediately, on the first day, and on the second day after delivery. Additionally, this study indicated a potential favorable effect on improving the integrity of the perineum. Nevertheless, this study did not find evidence supporting the effectiveness of perineal warm compresses in reducing the first-degree perineal lacerations and the rate of perineal lacerations requiring sutures.

### 4.2. Interpretation

In recent years, multiple evidence-based clinical practice guidelines [13,38,39] have recommended offering perineal warm compresses as a promising midwifery technique for perineal protection in the second stage of labor to reduce the incidence of perineal trauma and episiotomy and to increase the likelihood of an intact perineum. Healthcare professionals, including obstetrician-gynecologists, midwives, or nurse-midwives, as well as researchers with backgrounds in obstetric or midwifery care, are advised to discuss the benefits, risks and strategy with women. After obtaining informed consent, they can apply perineal warm compresses to women at the commencement of perineal stretching or during pushing [40]. It is crucial to stop if discomfort arises or at the women’s request. It is worth noting that most of the guidelines provide graded level A recommendations for perineal warm compresses to reduce severe perineal lacerations during the second stage of labor [13,38], indicating that the recommendations are based on relatively reliable research evidence. This study further supported the effects of perineal warm compresses on reducing the incidence of episiotomy and relieving the perineal pain within two days postpartum. These findings provide evidence to support a wider application in clinical practice. Guidelines also recommended developing local policies to standardize the procedure, such as ensuring a proper temperature of the warm packs to guarantee a safe application, documenting the implementation process in detail, and other organizational and management considerations [41]. The study also provides details on preparing a standardized process for perineal warm compresses. 

Numerous risk factors are associated with the occurrence of perineal trauma, including the first vaginal birth, maternal ethnicity (such as Southeast Asian race), maternal age over 35 years, a large birth weight or head circumference, malposition, a prolonged second stage of labor, and instrumental births [14,42,43,44,45]. Due to the physiological structure difference between nulliparous and multiparous women, perineal tears mostly occur in primiparous women; hence, special attention should be paid to this group. This study showed that the perineal warm compresses significantly reduced the occurrence of third- and/or fourth-degree perineal lacerations among primiparous women. This aligns with previous systematic reviews [28,31,46] and is consistent with the recommendations from RCOG [13] and the American College of Obstetricians and Gynecologists (ACOG) [38]. Perineal warm compresses also effectively decreased the incidence of episiotomy, in line with relevant studies [31,46]. Several possible mechanisms could explain these results. Firstly, warmth can dilate the blood vessels, promote blood circulation, and increase the pain threshold of the perineum [47]. Meanwhile, warm compresses can distract maternal attention and reduce psychological pressures related to pain. Secondly, based on the gate control theory of pain [48,49], excessive sensory stimulation like warmth sends impulses to the brainstem to block the pain gate from closing, reducing pain signals. Finally, warm compresses can strengthen the extensibility of the skin and the muscles, achieving the best perineal state and decreasing the occurrence of perineal lacerations, particularly in nulliparous women or those with thicker perineal tissues [50].

The evidence presented in this review indicated that perineal warm compresses were associated with short-term relief of perineal pain. These effects, particularly in reducing the perineal pain immediately after birth, align with the findings of a previous systematic review [46]. It is worth noting that the certainty of the aggregated results of the perineal pain relief immediately after birth were rated as “very low”. Therefore, further investigations are needed, and more high-quality RCTs are warranted to explore the effects of perineal warm compresses during childbirth. Perineal pain relief two days after birth obtained a “moderate” level in the certainty assessment, indicating that after receiving warm compresses, women are more likely to experience less prolonged pain due to improved perineal conditions. To enhance the positive maternal experience of childbirth, midwives and clinicians are advised to employ more evidence-based midwifery techniques.

The combined results failed to prove the beneficial effects of perineal warm compresses on reducing the first-degree perineal lacerations and the rate of perineal lacerations requiring sutures. This is consistent with the findings of the Cochrane systematic review [28]. Although the review conducted by Fadlalmola et al. [46] suggested that perineal warm compresses can decrease the incidence of perineal lacerations requiring sutures, such evidence was not confirmed in this review. However, it is worth noting that warm compresses have not been proven to be harmful and are widely accepted by midwives and doctors who use this technique to reduce the incidence of perineal laceration and to improve the perineum integrity of women [51,52,53]. Additionally, although one of the objectives of this systematic review was to explore the women’s feeling about receiving perineal warm compresses during the second stage of labor, the results did not provide data on this aspect. Therefore, future research, especially original studies, exploring women’s experience with perineal warm packs is encouraged to reveal the comprehensive effectiveness from both physical and psychological aspects among women.

### 4.3. Strengths and Limitations

In this study, we systematically searched, assessed, and analyzed RCTs relevant to perineal warm compresses during the second stage of labor, focusing specifically on primiparous maternal outcomes, especially those related to the perineum. Simultaneously, we employed the GRADEpro tool to evaluate the certainty of aggregated outcomes, leveling the evidence. This approach aims to provide relatively objective and reliable results for clinical midwifery practice. However, this review has some limitations. Firstly, although we developed a systematic search strategy to explore the relevant literature as comprehensively as possible, the restrictions to English or Chinese language may result in a potential publication bias. Furthermore, none of the pooled outcomes in this review were rated as “high” levels of evidence, suggesting that caution is needed when interpreting the results.

## 5. Conclusions

This systematic review and meta-analyses found that the perineal warm compresses during the second stage of labor significantly decreased the second-, third- and/or fourth-degree perineal lacerations, reduced the incidence of episiotomy, and provided short-term relief for perineal pain postpartum. Furthermore, there is a potential favorable effect on improving the integrity of the perineum with perineal warm compresses. However, the supportive effects of perineal warm compresses on reducing first-degree perineal lacerations and the rate of perineal lacerations requiring sutures have not been proven. More well-designed RCTs of perineal warm compresses during the second stage of labor can generate high-quality evidence for midwifery techniques, enabling evidence-based midwifery care for women. Future studies are needed to determine the effects on reducing perineal swelling, urinary incontinence postpartum, and the women’s experiences and feelings regarding perineal warm compresses.

## Figures and Tables

**Figure 1 healthcare-12-00702-f001:**
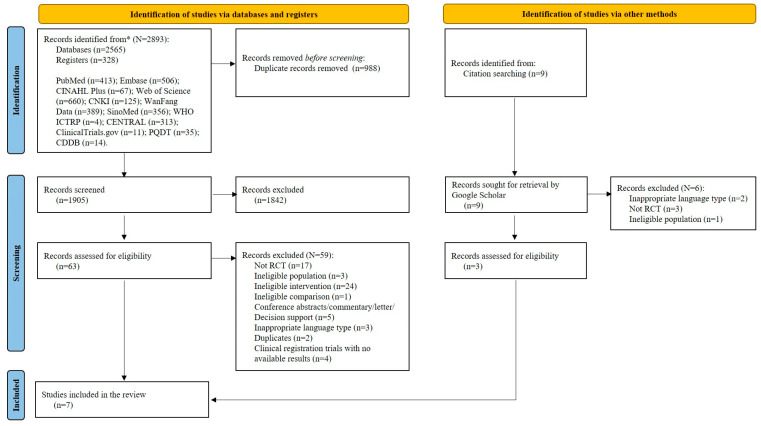
The PRISMA flowchart. **Notes:** * represents the total number of records identified from databases and registers.

**Figure 2 healthcare-12-00702-f002:**
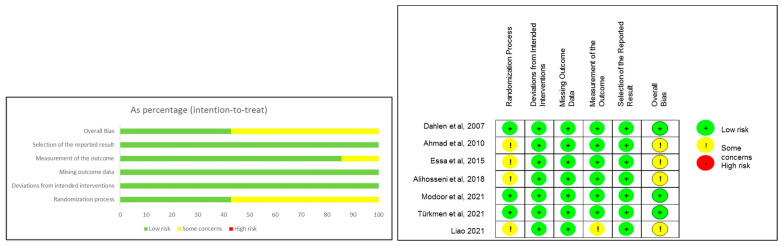
Summary of risk of bias for included studies [25,32,33,34,35,36,37].

**Figure 3 healthcare-12-00702-f003:**
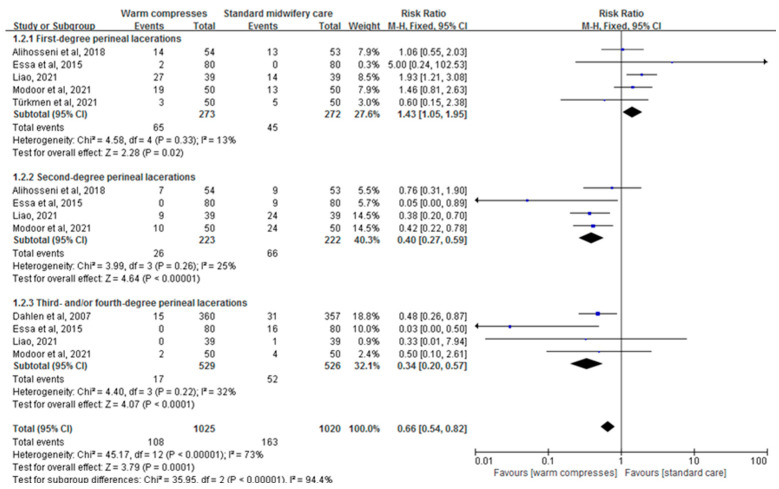
Effect of warm compresses on perineal lacerations [25,32,33,34,35,36,37].

**Figure 4 healthcare-12-00702-f004:**
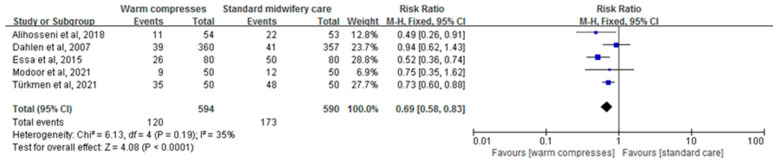
Effect of warm compresses on episiotomy [25,32,34,35,36].

**Figure 5 healthcare-12-00702-f005:**
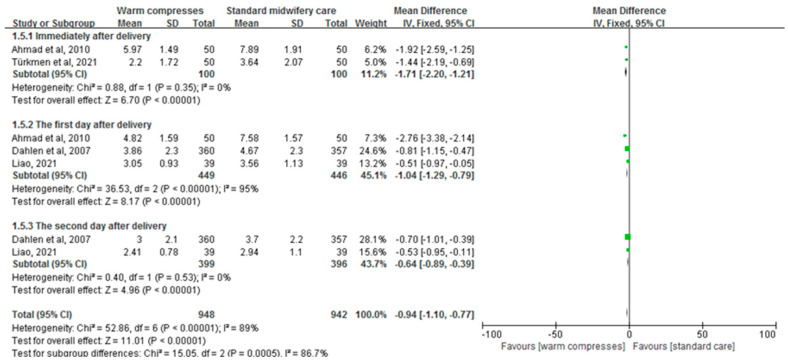
Effect of warm compresses on postpartum perineal pain [25,32,33,37].

**Table 1 healthcare-12-00702-t001:** Characteristics of the included studies (N = 7).

	Dahlen et al., 2007 [32]	Ahmad et al., 2010 [33]	Essa et al., 2015 [34]	Alihosseni et al., 2018 [35]	Modoor et al., 2021 [36]	Türkmen et al., 2021 [25]	Liao, 2021 [37]
Study location	Australia	Saudi, Arabia	Egypt	Iran	Saudi, Arabia	Turkey	China
Participant	Primiparous women	Primiparous women	Primiparous women	Primiparous women	Primiparous women	Primiparous women	Primiparous women
Size (I/C)	717 (360/357)	100 (50/50)	160 (80/80)	107 (54/53)	100 (50/50)	100 (50/50)	78 (39/39)
Intervention	Perineal warm compresses	Perineal warm compresses	Perineal warm compresses	Perineal heating pad	Perineal warm compresses	Perineal warm compresses	Perineal warm compresses
Performer	Midwives	Researcher	Researcher	Midwives	Researcher	Researcher	Researcher
Time to start	When the baby’s head began to distend the perineum and the woman felt perineal extension	During the whole second stage of labor	NR	The start of the second stage of labor	In the second stage of labor	In the second stage of labor after women were taken to the obstetrical table	When the fetal head is exposed and the mother feels perineal extension
Duration packs were held to perineum every time	During contractions until delivery	During each uterine contraction for 10 min	During each uterine contraction	Continuously during the second stage of labor	Continuously during the second stage of labor	5–15 min with no interruption	5–6 min
Temperature of the warm packs	38–44 °C	NR	38–44 °C	50 °C continuously	NR	NR	NR
Water temperature of the jug	45–59 °C	45° C	45–49 °C	NR	45–49 °C	45–50 °C	45–48 °C thermostat
Time to keep packs in water	Resoaked between uterine contractions	Resoaked between uterine contractions	Resoaked between uterine contractions	NR	NR	NR	NR
Frequency to replace water in the jug	Every 15 min	Every 15 min	Every 15 min or if the temperature dropped below 45 °C	NR	NR	NR	NR
Time to stop	Until delivery	Until delivery of the fetal head	During the whole second stage of labor until delivery	During the whole second stage of labor until delivery	Until the fetal head is crowned	After 5–15 minutes’ intervention	Until the fetal head is crowned
Comparison	Standard midwifery care	Standard midwifery care	Standard midwifery care	Standard midwifery care	Standard midwifery care	Standard midwifery care	Standard midwifery care
Outcome 1	Perineal pain	Perineal pain	Perineal pain/Behavioral pain	NR	Perineal pain	Perineal pain	Perineal pain
Measurement	VAS	VAS	NRS/BPS	NR	NRS	VAS	NRS
Data Collection time point	Three times: immediately after delivery; on the first day after delivery; and on the second day after delivery	Three times: before intervention; immediately after delivery; and on the first day after delivery	Twice: before intervention and immediately after delivery	NR	NR	Four times: before intervention; 10 min after intervention; immediately after delivery; and 2 h after delivery	Twice: on the first day after delivery and on the second day after delivery
Appraiser	Women themselves	Women themselves	Women themselves and the researcher	NR	Women themselves	Women themselves	Women themselves
Outcome 2	Perineal outcomes	NR	Perineal outcomes	Perineal outcomes	Perineal outcomes	Perineal outcomes	Perineal outcomes
Measurement	(1) Intact perineum; (2) Perineal laceration: first-degree; second-degree; third-degree; and fourth-degree (3) Episiotomy (4) Suture need for perinea	NR	(1) Intact perineum; (2) Perineal laceration: first-degree; second-degree; third-degree; and fourth-degree (3) Episiotomy (4) Suture need for perinea	(1) Intact perineum; (2) Perineal laceration: first-degree; second-degree; third-degree; and fourth-degree (3) Episiotomy (4) Suture need for perinea	(1) Intact perineum; (2) Perineal laceration: first-degree; second-degree; third-degree; and fourth-degree (3) Episiotomy (4) Suture need for perinea	(1) Intact perineum; (2) Perineal laceration: first-degree; second-degree; third-degree; and fourth-degree (3) Episiotomy (4) Suture need for perinea	(1) Intact perineum; (2) Perineal laceration: first-degree; second-degree; third-degree; and fourth-degree
Data Collection time point	Immediately after delivery	NR	Immediately after delivery	Immediately after delivery	Immediately after delivery	Immediately after delivery	Immediately after delivery
Appraiser	Researcher	Researcher	Researcher	The evaluator was blinded	Researcher	Researcher	Researcher
Outcome 3	Urinary incontinence	NR	Need for pain relief	NR	NR	The comfort of the women	Perineal swelling
Measurement	By telephone	NR	Assessment tool	NR	NR	Postpartum Comfort Scale	By the soft ruler
Data Collection time point	At 6 weeks postpartum and 3 months postpartum	NR	During the second stage of labor	NR	NR	2 h after delivery	Immediately after delivery
Appraiser	Researcher	NR	Researcher	NR	NR	Researcher	Midwives

VAS: Visual Analog Scale; NRS: Numerical Rating Scale; BPS: Behavioral Pain Scale; NR: not reported.

## Data Availability

The data that support the findings of this study are available on the Appendix A.

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
