# Peer review of "Effects of Perineal Warm Compresses during the Second Stage of Labor on Reducing Perineal Trauma and Relieving Postpartum Perineal Pain in Primiparous Women: A Systematic Review and Meta-Analyses"

_healthcare, 2024, doi:10.3390/healthcare12070702_

Round 1

Reviewer 1 Report

Comments and Suggestions for Authors

Dear author’s 

I was pleased to review your interesting article and i have the following comment’s:

The article is very well written but do you think that this technique is fesable In developed countries? It is very difficult to perform this in a hospital, in my opinion. Do you use this technique in your center?

Outhewise the article is very well written.

Author Response

Response: Thank you for your time. Based on our findings, perineal warm compresses, by placing warm packs in the perineum during the second stage of labor, can effectively improve the perineal integrity and decrease perineal pain postpartum. In the present, some first-class hospitals in our country have already implemented it as an evidence-based midwifery technique during the second stage of labor. According to the research and the interview, both professional staff and women who received this technique had high acceptances of perineal warm compresses. In particular, being non-invasive and safe to conduct, perineal warm compresses were favored by many midwives and women for relieving perineal pain and increasing women’s comfort. However, due to the lack of midwifery staff, it has not been widely carried out in China. Expanding the application of the evidence-based midwifery techniques including perineal warm compresses through the implementation science methods and framework is the direction in the future.

Reviewer 2 Report

Comments and Suggestions for Authors

Dear authors,

It was my pleasure to review this manuscript that attempts to study the effects of perineal warm compresses during the second stage of labor to reduce perineal trauma and relieve postpartum perineal pain in primiparous women through a systematic review and meta-analysis.

First of all I would like to ask the authors why only primiparous women were included and not all those who had a normal birth.

I found the manuscript very interesting and well structured. The introduction, methodology and results seemed correct to me. In the discussion section, I would suggest stating the strengths and limitations in the final part, after the interpretation of section (4.3).

Secondly, I suggest reviewing the style of bibliographical references. According to Vancouver style, after each surname there should not be a “,” and after initials there should not be a “;”. Furthermore, in all references the volume is indicated, but none of them indicate the number, which must be in parentheses.

Otherwise, I find the manuscript very interesting to be published.

Thank you

Kind regards

Author Response

Comments 1: First of all I would like to ask the authors why only primiparous women were included and not all those who had a normal birth.

Response 1: Thank you for your time. We did not include all women who had a normal birth because in the clinical practice, the primiparas are more likely to suffer perineal laceration than multiparas. According to the literature research, the rate of perineal trauma is typically higher in women who give birth vaginally for the first time. In a normal delivery, the incidence of natural perineal laceration and the rate of episiotomy among nulliparous women was 3 times and 2.5 times higher than among parous women, respectively. And nulliparous women are a high-risk group for severe perineal tears, the risk of which in nulliparous women increased 5.32 times comparing to multiparous women. Therefore, the first birth is widely recognized as a remarkable risk factor for perineal injury. We added this part of the description to clarify this issue in the introduction section, please see page 1, line 33-36, line 65-66, highlighted.

Comments 2: I found the manuscript very interesting and well structured. The introduction, methodology and results seemed correct to me. In the discussion section, I would suggest stating the strengths and limitations in the final part, after the interpretation of section (4.3).

Response 2: Agree. We have accordingly changed the order. Thank you for your kind advice, we adjusted the order of these two parts in the discussion section, please see page 3 and page 5, highlighted.

Comments 3: Secondly, I suggest reviewing the style of bibliographical references. According to Vancouver style, after each surname there should not be a “,” and after initials there should not be a “;”. Furthermore, in all references the volume is indicated, but none of them indicate the number, which must be in parentheses. Otherwise, I find the manuscript very interesting to be published. Thank you. Kind regards.

Response 3: Thank you for your reminder. As for the format of the references, we reread the “MDPI Reference List and Citations Style Guide”, which is a little bit different from the Vancouver style.

Reviewer 3 Report

Comments and Suggestions for Authors

Dear Authors,

I was glad to get involved in the review process for the article titled Effects of perineal warm compresses during the second stage of labor on reducing perineal trauma and relieving postpartum perineal pain in primiparous women: a systematic review and meta-analyses .

The article has substantial strengths which deserve to be stressed: it is original and centered around a highly relevant area of OB/GYN research; it is well-centered in its objective and potentially appealing to a relatively wide readership; it is competently structured (although it is a somewhat unusual structural framework for a scientific article, with way too many subchapters which can come across as confusing) and relies on sound methodology, as far as I was able to determine; the tables and figures are quite well-crafted and effective at conveying key data and findings.

I do believe however that the Discussion falls short to a degree, and the manuscript will benefit from greater breadth and contextualization as far as delivery risk factors are concerned, and how perineal warm compresses can play a role against such a backdrop. In order to make the most out of its premise and findings, the article needs to be more comprehensive. Defining the role of guidelines and evidence-based recommendations (which have a medicolegal value within the framework of proper patient care during delivery and call for documentable compliance on the part of healthcare professionals) is necessary to add an extra element of depth to the discussion. In fact, fleshing out such fundamental elements would highlight the importance of the article's core findings and conclusions, therefore I recommend they be developed and succinctly expounded upon in the Discussion section. 

The following sources should be drawn upon and cited:

doi: 10.1097/OGX.0000000000001225. 

doi: 10.1080/14767058.2017.1281243. 

Though the article is fairly well-written overall I would still have it go through further proofreading by a native speaker of English, in order to iron out a few inconsistencies and less-than-ideal grammar/syntax.

Overall, I believe that the article and its peculiarities can be a valuable contribution to a highly relevant area of research.

Best regards.

Comments on the Quality of English Language

Though the article is fairy well-written overall I would still have it go through further proofreading by a native speaker of English, in order to iron out a few inconsistencies and less-than-ideal grammar/syntax.

Author Response

Comments 1: The article has substantial strengths which deserve to be stressed: it is original and centered around a highly relevant area of OB/GYN research; it is well-centered in its objective and potentially appealing to a relatively wide readership; it is competently structured (although it is a somewhat unusual structural framework for a scientific article, with way too many subchapters which can come across as confusing) and relies on sound methodology, as far as I was able to determine; the tables and figures are quite well-crafted and effective at conveying key data and findings.

Response 1: Thank you for your time. We really appreciated your kind comments.

Comments 2: I do believe however that the Discussion falls short to a degree, and the manuscript will benefit from greater breadth and contextualization as far as delivery risk factors are concerned, and how perineal warm compresses can play a role against such a backdrop. In order to make the most out of its premise and findings, the article needs to be more comprehensive. Defining the role of guidelines and evidence-based recommendations (which have a medicolegal value within the framework of proper patient care during delivery and call for documentable compliance on the part of healthcare professionals) is necessary to add an extra element of depth to the discussion. In fact, fleshing out such fundamental elements would highlight the importance of the article's core findings and conclusions, therefore I recommend they be developed and succinctly expounded upon in the Discussion section. The following sources should be drawn upon and cited: doi: 10.1097/OGX.0000000000001225. doi: 10.1080/14767058.2017.1281243.

Response 2: Agree. We have accordingly revised the discussion to emphasize these points you suggested. Thank you very much for your guidance on the issue of the insufficient discussion. As you suggested, we sorted out and supplemented the following in-depth discussion from 3 aspects. (1) we addressed the role of guidelines and evidence-based recommendations to clarify the importance of the perineal warm compresses; (2) we have compounded the delivery risk factors relevant to the perineal trauma during the natural delivery; (3) we explained the mechanism and significance of how the perineal warm compresses can reduce the perineal injury and decrease the perineal pain after delivery. We also supplemented the corresponding bibliographical references, including the two articles you mentioned. Pease see page 3-4, highlighted.

Comments 3: Though the article is fairly well-written overall I would still have it go through further proofreading by a native speaker of English, in order to iron out a few inconsistencies and less-than-ideal grammar/syntax. Overall, I believe that the article and its peculiarities can be a valuable contribution to a highly relevant area of research.

Response 3: As for the English writing issue, thank you for your suggestion, we had our manuscript checked and proofread by a peer colleague fluent in English writing.

4. Response to Comments on the Quality of English Language

Point 1: Though the article is fairy well-written overall I would still have it go through further proofreading by a native speaker of English, in order to iron out a few inconsistencies and less-than-ideal grammar/syntax.

Response 1: We had our manuscript checked and proofread by a peer colleague fluent in English writing.

Round 2

Reviewer 3 Report

Comments and Suggestions for Authors

Dear Authors,

I can certainly appreciate the extent to which you have succeeded in improving your manuscript, which now reads substantially more comprehensive and well-rounded overall.

In light of its strengths, relevance and novelty in a highly meaningful area of ob/gyn research, I am recommending approval for publication.

Best regards.

Comments on the Quality of English Language

Language has improved substantially and now lives up to scientific standards of quality.